# A Chewable Cure “Kanna”: Biological and Pharmaceutical Properties of *Sceletium tortuosum*

**DOI:** 10.3390/molecules26092557

**Published:** 2021-04-28

**Authors:** Madira Coutlyne Manganyi, Cornelius Carlos Bezuidenhout, Thierry Regnier, Collins Njie Ateba

**Affiliations:** 1Department of Biological and Environmental Sciences, Faculty of Natural Sciences, Walter Sisulu University, PBX1, Mthatha 5117, South Africa; 2Unit for Environmental Sciences and Management, Faculty of Natural and Agricultural Sciences, North-West University, Private Bag X6001, Potchefstroom 2520, South Africa; Carlos.Bezuidenhout@nwu.ac.za; 3Department of Biotechnology and Food Technology, Tshwane University of Technology, Pretoria 0001, South Africa; regniert@tut.ac.za; 4Food Security and Safety Niche Area, Faculty of Natural and Agricultural Sciences, North-West University, Mmabatho, Mafikeng 2735, South Africa; Collins.Ateba@nwu.ac.za

**Keywords:** *Sceletium tortuosum*, kougoed, well-being, biological properties, bioactive compounds

## Abstract

*Sceletium tortuosum* (L.) N.E.Br. (Mesembryanthemaceae), commonly known as kanna or kougoed, is an effective indigenous medicinal plant in South Africa, specifically to the native San and Khoikhoi tribes. Today, the plant has gained strong global attraction and reputation due to its capabilities to promote a sense of well-being by relieving stress with calming effects. Historically, the plant was used by native San hunter-gatherers and Khoi people to quench their thirst, fight fatigue and for healing, social, and spiritual purposes. Various studies have revealed that extracts of the plant have numerous biological properties and isolated alkaloids of *Sceletium tortuosum* are currently being used as dietary supplements for medicinal purposes and food. Furthermore, current research has focused on the commercialization of the plant because of its treatment in clinical anxiety and depression, psychological and psychiatric disorders, improving mood, promoting relaxation and happiness. In addition, several studies have focused on the isolation and characterization of various beneficial bioactive compounds including alkaloids from the *Sceletium tortuosum* plant. Sceletium was reviewed more than a decade ago and new evidence has been published since 2008, substantiating an update on this South African botanical asset. Thus, this review provides an extensive overview of the biological and pharmaceutical properties of *Sceletium tortuosum* as well as the bioactive compounds with an emphasis on antimicrobial, anti-inflammatory, anti-oxidant, antidepressant, anxiolytic, and other significant biological effects. There is a need to critically evaluate the bioactivities and responsible bioactive compounds, which might assist in reinforcing and confirming the significant role of kanna in the promotion of healthy well-being in these stressful times.

## 1. Introduction

In a developing country such as South Africa, a feasible dual health care system is practiced by incorporating current Western medical practice with traditional medical health care. Approximately 80% of the world’s population and 52% of South Africans, especially blacks, use traditional medicine and practices for primary health care. In addition to the fact that it is part of their cultural heritage, the traditional health care system provides an affordable, personalized, and culturally accepted alternative to the costly modern clinical system. The South African health care system is overwhelmed by the private and public health care system, however, the ratio of traditional healers to allopathic doctors is estimated at 10 to 1 [1,2,3].

Indigenous medicinal plants have been a key resource used for centuries in various native tribes around the world. Most South Africans use plants to treat physical and psychological illnesses/issues [4]. Furthermore, South Africa is rich in traditional healing methods and diverse fauna and flora, with approximately 30,000 flowering plant species, which account for 10% of the world’s higher plant species. There has been a universal trend toward the use of medicinal plants for various human diseases and aliments for social and economic benefits [5]. Bioactive compounds such as alkaloids, phenolic, flavonoids, tannins, glycosides, saponins, and terpenoids have been isolated from medical plants [6,7].

*Sceletium tortuosum* is no expectation, as extensive research has been conducted on the chemistry of *S. tortuosum* alkaloids [8] as the plant contains a large profile of alkaloids such as mesembrine, mesembrenone, mesembrenol, tortuosamine, and chennaine, and alkaloids have an effect on a number of central nervous system targets. For example, an ethanolic extract of *S. tortuosum* with purified alkaloids mesembrine, mesembrenol, and mesembrenone showed inhibitory effects on serotonin (5HT) reuptake and phosphodiesterase4 (PDE4) activity [9]. *S. tortuosum* is a succulent, flowering plant casually known as kougoed or kanna, indigenous to South Africa [9]. The plant is a member of the Mesembryanthemaceae family [10]. In addition, *Sceletium* is well-known as “Kanna, Channa, and Kougoed”, meaning something to chew or is chewable. The plant is traditionally known for its ability to elevate mood, reduce stress, tension, anti-anxiety and its tranquilizing properties [10].

Furthermore, it is used for illnesses such as abdominal pains, toothache, and some people chew, smoke, or use it as tea or snuff mostly for pressure. The antidepressant and anxiolytic clinical effects of *S. tortuosum* have been found both in case reports [11] and more recently, double-blind studies [9]. Anecdotal records reveal that the Khoikhoi and San people have used this plant since ancient times as an essential part of the indigenous culture and materia mediac. Hunter gatherers and pastoralists use *S. tortuosum* for the endurance of hunting attacks and management of stress that comes with living in dry and challenging environments of Bushman land, Namaqualand, and the Karoo [12].

Bennett and colleagues [13] reported that *S. tortuosum* showed potent anti-inflammatory capacity in the context of chronic disease. Furthermore, high levels of mesembrine extracted from *S. tortuosum* displayed potential cytoprotective and mild anti-inflammatory properties in the setting of acute inflammation in the peripheral compartment. In addition, it has also been proved to target specific enzymes in the adrenal cortical steroid synthesis pathway and reduce glucocorticoid synthesis. In terms of diabetes and obesity, this is significant since the etiology of both conditions is linked to chronically elevated pro-inflammatory cytokine and glucocorticoid levels [13].

*S. tortuosum* including other species have become attractive commodities in the commercialization of South African medicinal plants. Various forms of the plant are currently being sold, for example, tea bags, often mixed with Red Bush Tea (Aspalathus linearis), or Honeybush tea (*Cyclopia* spp.), are purchased in South African supermarkets. Extracts of the plant are accessible in raw powdered plant material, tablets, and capsules, which are frequently traded over the Internet and individuals use it to improve their sense of well-being and reduce stress [8]. Furthermore, *S. tortuosum* plants provide an effective, efficacious, and affordable natural treatment for veterinary and pharmaceutical purposes. Historic reports have shown that Sceletium plants are culturally used by traditional healers for psychological, spiritual, and medical functions [1]. Thus, the purpose of this review was to establish a brief historical overview, origin, phytochemistry as well as a comprehensive outlook in recent pharmacological, veterinary, and medicinal advances with regard to a chewable South African succulent genus, Sceletium.

## 2. Methodology Framework

Research studies/papers were collected using a wide range of database platforms such as Scopus, PubMed, Google Scholar, Web of Science, and ScienceDirect. Keywords such as *Sceletium tortuosum*, kougoed, well-being, and biological properties were used to search for relevant studies. A total of 1040 scientific papers were retrieved from Google Scholar and a minimum of 38 from PubMed. Due to the limited number of studies, there was no restriction on the year of publication. However, our search focused on in vitro culture studies, animal studies as well as human investigations. Thus, the studies were linked back to the traditional uses of the plant for therapeutic benefits. Abstracts were primarily screened before reviewing the full publications.

## 3. History, Description and Distribution of *Sceletium*

The oldest written evidence of *Sceletium* has been found on a painting done by Simon van der Stel in 1685, approximately 435 years old (Figure 1a). According to historical records, the plant is perennial, short-lived with creeping stems and overlapping pairs of leaves that have glistening water cells (bladder cell idioblasts) on their surfaces. In 1662, the Dutch colonial administrator, Jan van Riebeeck (Figure 1b) traded with native tribes in Southern Africa, exchanging sheep for ‘kanna’. It was identified as a ginseng-like herb by the European population. However, prior to this, South African pastoralists and hunter-gatherers had been using the plant as a mood-altering substance from prehistoric times [1,14,15].

*S. tortuosum* is a small succulent plant with trimmed branches that thicken along and become slightly woody with age. The succulent plant consists of water cells. These are visible on the leaves with curved tips entailing three to five main veins. *Sceletium* has a climbing pattern and the succulent leaves have “bladder cells” or idioblasts. Furthermore, the dicotyledonous flowering part of the plant comprises whitish to pale yellow, sometimes, pale pink petals supported by pedicellate (Figure 2b). The calyx contains four or five sepals. Imbricate leaves with incurved tips represent a very distinct characteristic, keeping in mind that in its hygrochastic state, the fruit opens up. The *Sceletium* genera belongs to the family Aizoaceae and subfamily Mesembryanthemoideae, recognized in 1925 by N.E. Brown. When the leaves are dry, *S. tortuosum* has a skeleton-like structure (skeletonized).

Hence, the botanical name, *Sceletium,* comes from the Latin word “sceletus”, due to the prominent leaf veins. *Sceletium* genera consists of eight species, known for their persistent dry leaves that become skeletonized (Figure 2a) [1,17]. The *Sceletium* genera, although native to southwestern parts of South Africa, has attracted global attention since it enhances a sense of well-being and healing properties for anxiety, depression, and stressed individuals. Its geographical distribution is from the Namaqualand arid region of Namibia to Montagu in the Western Cape Province of South Africa, near the Western Little Karoo through to Aberdeen of the Eastern Cape Province of South Africa. *Sceletium* genera are succulent plants that flourish in sandy loam soil in predominantly dry environments under shrubs in partial shade. In the dry season, the leaves dry out, enclosing young leaves to protect them against unfavorable environmental conditions. In addition, they propagate extremely well in rockeries and pots [18,19].

## 4. Phytochemical Constituents of *Sceletium*

In 1898, Meiring was one of the first to study the phytochemical components of the *S. tortuosum* crude alkaloid mixture [1]. Historic investigation by Zwicky (1914) [20] also proved that various alkaloids such as mesembrine and mesembrenine were isolated from *S. tortuosum* and *S. expansum. Sceletium* alkaloids was further studied for the extraction and synthesis of mesembrine, mesembrenine (=mesembrenone), mesembrinol (=mesembranol), and the possible artifact chinnamine [21]. Recently, several studies have shown high concentrations of alkaloid, specifically mesembrine as well as other *Sceletium* alkaloids extracted from *Sceletium* plants [22]. Expert research has confirmed that the principal active component in *S. tortuosum* is mesembrine [23]. In-depth research established that genus *Sceletium* produces indole alkaloids, for instance, mesembrenol, mesembranol, mesembrine, and mesembranone (Figure 3), which has a chemical formulae designated in U.S. Pat. No. 6288104.

Further research revealed that U.S. Pat. No. 6288104 mesembrine was the only alkaloid detected in the leaves of *S. tortuosum* [24]. Patnala and Kanfer [25] performed high-performance liquid chromatography (HPLC) with UV, coupled with online mass spectroscopy, using six investigated *Sceletium* plants including *S. tortuosum.* Chemical analysis showed that mesembrenone and mesembrine were present in high levels, however, mesembranol and epimesembranol were detected in low levels in *S. expansum. S. strictum* eluded several alkaloids such as mesembrine, mesembrenone, and either 4′-0-demethylmesembrenone or 4′-0-demethylmesembrenol [25].

Shikanga and co-workers [26] conducted a study on 151 wild *S. tortuosum* plants obtained from 31 areas in the Western Cape Province of South Africa. The researchers used gas chromatography-mass spectrometry (GC-MS) analysis in the chemical investigation of acid/base extracts. The results showed unpredictability in the content of mesembrine-type alkaloid [26]. Zhaoa and colleagues [27] used proton nuclear magnetic resonance (1H-NMR) spectroscopy of methanol extracts and ultra-performance liquid chromatography-mass spectrometry (UPLC-MS) of acid/base extracts to determine the chemotypic distinctions in *S. tortuosum* in two locations in South Africa. Promising findings have been reported for pinitol, *Sceletium* alkaloids, and two alkylamines discovered for the first time from *S. tortuosum* [27]. *S. tortuosum* is a miracle and mood-elevating plant used for the treatment of anxiolytic agents, without disregarding its different antimicrobial properties [28].

## 5. Biological and Pharmaceutical Properties of *Sceletium* sp.

### 5.1. Antimicrobial Properties of Sceletium Plants

In light of the ongoing research on alkaloids isolated from *Sceletium* plants, several studies have reported the biological and pharmacological potentials of *Sceletium* plants [1,30,31]. Traditional preparation of the plants through fermentation, referred to as “kougoed”, has renewed attention since it was reported that the fermented form produces higher levels of alkaloids than the dried the plant material [32]. However, there is antagonistic or synergistic interactions of phytochemical compounds within the plant [33,34]. Traditional practices revealed that Sceletium plants have abilities for elevating mood and reducing stress and tension, of anti-anxiety and tranquilizing properties [35], and are used for the treatment of abdominal pain and toothache. Additionally, some people chew, smoke, drink as tea, or snuff it, mostly to relieve pressure [15]. Nowadays, the pharmacological prospective of *Sceletium* plants has been explored and established using various scientific methodologies such as in vitro and in vivo animal models, coupled with human studies.

Kapewangolo and co-workers [36] investigated commercialized crude extracts of *S. tortuosum* for anti-HIV as well as free radical scavenging potential. Furthermore, a phytochemical analysis was conducted. The results revealed that *S. tortuosum* extracts contained anthraquinones, polyphenols, terpenes, anthocyanin, alkaloids, tannins, glycosides, coumarins, and carbohydrates. A stronger anti-HIV activity was reported by ethanol and ethyl acetate extracts of *S. tortuosum*, which displayed HIV-1 RT and PR, respectively. HIV-1 RT data showed IC50 values of <50 and 121.7 ± 2.5 μg/mL for ethanol and ethyl acetate extracts, respectively. The results also showed that the plant is a rich source of phytochemicals, with antioxidant activity. The authors concluded that *S. tortuosum* provides novel leading bio-compounds possessing beneficial features such as new anti-HIV and radical scavenging bioactive compounds [36].

For centuries, *S. tortuosum* has been traditionally used as an antimicrobial agent, namely; as a painkiller for headache, local anesthetic action, abdominal pain, and for the respiratory tract [1,23]. The native people of South Africa would chew the plant to relieve toothache. Several studies have revealed a wide spectrum of therapeutic potentials of *S. tortuosum*. Many studies have shown that the antimicrobial properties of medicinal plants are due to bioactive compounds, either a single or combined effect. *Sceletium* plants have been reported to be an excellent source of alkaloids, specifically mesembrine alkaloids. Moreover, mesembrenone alkaloids have been reported to exhibit antimicrobial properties [22,23,24]. Although few studies have been conducted, this review provides direction for future studies on *S. tortuosum*.

### 5.2. Anti-Stress Properties of Sceletium Plants

Stress is a nonspecific physical/psychological condition that results in mental tension or physiological responses that might lead to sickness. Moreover, prolonged life stress is considered a serious life-threatening condition and might result in chronic physical and mental disease [37]. Stress plays a key role in the states of various diseases such as hypertension, diabetes, peptic ulcer, immuno-suppression, anxiety reproductive dysfunctions, central nervous system (CNS), metabolic system, and endocrine system [38,39]. With regard to the above, stress has been associated with mood, cognition, schizophrenia, and other psychiatric disorders [40,41]. According to the Global Organization for Stress, in approximately 77% of individuals who are stressed, their condition impacts on their physical health. Stress affects the mental health of 73% of individuals and disrupts the sleeping habits of 48% of people. In addition, 33% of individuals experienced extreme stress [42]. Various formulae of *Sceletium* products such as dried plant material, tinctures, teas, tablets, and capsules have been used to enhance well-being and act as a stress reliever. Anecdotal, historical, and traditional records support the above statement, with scientific research data [43]. Current research emphasizes the use of medicinal plants as a pharmacological anti-stress treatment to combat functional, behavioral, and molecular issues caused by stress as a suitable clinical solution [44,45,46].

Solati et al. [47] conducted an in-depth review on the effects and mechanisms of medicinal plants on stress hormones (cortisol). The results showed that *Sceletium* is one of the potential sources of an anti-stress agent [47]. The researchers employed the in vivo model of psychological stress using male Wistar rats in a double study. Placebo or 5 or 20 mg/kg/day of *S. tortuosum* extract every day for 17 days through oral administration and half of the test rats were subjected to moderate stress not exceeding 1 h for the last three days of treatment. Behavior assessments showed reduced stress on lower dose *Sceletium.* Through self-soothing behavior, further analysis reduced stress-induced corticosterone concentrations [10]. In human studies, *Sceletium* plant extract (Zembrin) was used to establish acute effects in a placebo double-blind, cross-over study. The study involved 16 healthy patients in a perceptual-load and an emotion-matching task. The first report showed the effectiveness of *S. tortuosum* on the threat circuitry of the human brain. This showed that the dual 5-HT reuptake and PDE4 effects had anxiolytic activity by attenuating the subcortical threat swiftly [48]. Bennett et al. [49] further showed the anti-stress effectiveness of *Sceletium* alkaloids, which affected the central enzymes (MAO-A) thus, preventing the adrenal steroidogenesis by blocking the CYP17, 3βHSD, and 17βHSD. With regard to the above, this showed the tremendous potential of the *Sceletium* plant as an anti-stress agent [49]. Stress and depression have a strong association with each other and can lead to serious life-threatening outcomes. There is growing interest in moving back to natural alternatives to combat various diseases, conditions, and aliments [50]. Traditional medicinal plants, particularly *S. tortuosum*, have been used for centuries as anti-stress, anti-anxiety, and anti-depression medications.

*S. tortuosum* has been widely known to promote a sense of well-being and relieve stress. The native hunter-gatherers and herders chewed raw *S. tortuosum* leaves to cope with life stress [36]. Nowadays, *S. tortuosum* is commercialized as a dietary supplement, stress reliever, and natural health medicines [27,48]. With an ever-growing stressful way of life, people are seeking natural alternatives to aid in achieving a sense of well-being.

### 5.3. Anti-Depressant Properties of Sceletium Plants

According to the World Health Organization (WHO), depression is a serious mental disorder or illness affecting over 264 million individuals globally. Depression can result in suicide, if not treated. Suicide was reported to be the second leading cause of fatalities in people aged 15–29 years. Moreover, approximately 800,000 individuals commit suicide per annum. Currently, therapeutic procedures for treating depression include psychological and pharmacological treatments for moderate and severe depression. In addition, 76% and 85% of patients in low and middle-income countries are not receiving treatments for mental disorders, respectively [51]. In response, traditional historic and anecdotal records show the effectiveness of medicinal plants where the plants were used for pharmacological treatment of mental health problems and disorders, particularly depression [52,53].

Among these medicinal plants, succulent *S. tortuosum* is widely known for promoting a sense of well-being. The plant improves mood, thus has anti-depression properties [15,26,29,54]. It has been found that compounds such as mesembrine, mesembrenol, and mesembranone are the key players. They have potent 5-HT as described by U.S. Pat. No. 6288104, which has an uptake inhibitory activity. Furthermore, it has been very active on mental health patients (mild to moderate depression). The compound has been stated in the past to be a frail PDE4 inhibitor [55]. Pervez and colleagues [56] conducted an in-depth study on plant alkaloids as an alternative treatment for depression in the last decade. The researchers established that *S. tortuosum* has an abundance of various alkaloids used in clinically depressed people [56,57].

Shahrajabian et al. [58] argued that during epidemics or pandemics, people experience serious mental health problems such as anxiety, traumatic stress, and depressive signs. In addition, the use and administration of traditional herbal medicines will assist individuals. *S. tortuosum* has been reported to treat depression, stress, and anxiety during these difficult times [58]. A study was conducted on the mesembrine of *S. tortuosum*, used as a substitute treatment for depression. Succulent *S. tortuosum* was obtained from the Cape region of South Africa. The results showed that *Sceletium* extracts have various alkaloids mesembrines, which have a calming effect and anxiolytic properties. The alkaloid family provides either serotonin reuptake inhibitors (SRI) or phosphodiesterase-4 (PDE4) inhibitors. In an animal model, a low concentration of (10 mg/kg) and a high dose (80 mg/kg) of *S. tortuosum* extracts were administered to BALB/c mice. Data further showed that pharmacological analyses displayed mesembrine alkaloids in *S. tortuosum* and possessed anti-depressant activity [59]. Many studies have revealed *S. tortuosum* as an anti-depressant agent. These properties have been articulated in research papers using different models such as in vitro studies, animal models [59] as well as human studies. The fact that this plant is used as a stress reliever, mood elevator, and is a sedative means that it is an ideal anti-depressant candidate [54].

### 5.4. Anxiolytic Properties of Sceletium Plants

Anxiety disorders are mental health conditions, considered as feelings of worry. They contribute to disability worldwide relating to other common conditions including phobias, post-traumatic syndrome, postmenopausal stress, cognitive dysfunction, and somatization. Individuals suffering from anxiety demonstrate functional impairment and the tendency to develop comorbid psychiatric disorders [60]. Sharma and co-workers [60] identified *S. tortuosum* as one of the most potent medicinal plants used globally for the treatment of anxiety-associated conditions and anxiety. Pharmacological and clinical investigations revealed that medicinal plants including *S. tortuosum* exhibit anxiolytic properties [60]. A zebrafish model study was conducted on ethnopharmacological experiments to determine anxiolytic properties [60].

A total of 28 plant extracts were subjected to the maximum tolerated Concentration (MTC) assay. The results showed an improvement in anxiolytic activity in the zebrafish model. Moreover, *S. tortuosum* displayed an anti-anxiety effect in zebrafish larvae. This is part of a full in vivo endorsement of the traditional use of the plant [61]. In current research, gamma-aminobutyric acid (GABA) and δ-opioid receptor are the key players in depression and anxiety conditions [62,63,64]. Several studies have shown mesembrine isolated from *S. tortuosum* plants. The plant had agonist actions on GABAA, µ-opioid, δ2-opioid, cholecystokinin-1, and melatonin-1 E4-prostaglandin receptors. This might be responsible for the anti-anxiolytic properties in vivo animal models [65,66,67]. A recent clinical study was conducted using plant extracts of *S. tortuosum* (trademarked―Zembrin^®^, Morristown, NJ, USA) to determine the anti-anxiolytic potential of the plant. The study was placebo-controlled, double-blind whereby 20 young healthy volunteers were administrated a single dose of *S. tortuosum* (25 mg, Zembrin^®^). Prior to this, participants were placed under stressful conditions to invoke feelings of stress and or anxiety.

The results showed levels of anxiety were inferior in the Zembrin^®^ group compared to the placebo group. Behavioral evidence demonstrated the efficacy of *S. tortuosum* (25 mg Zembrin^®^) as an anti-anxiolytic agent [68]. Fountain [69] conducted a preclinical model assay on chicks to determine the effectiveness against anxiety and depression of *S. tortuosum* plant extracts. Male Silver Laced Wyandotte chicks aged 4–6 days were exposed to stress. This resulted in depression-anxiety-like conditions. *S. tortuosum* extracts (10, 20, 30, 50, 75, or 100 mg/kg) were administrated to the chicks prior to stress. The results suggest that initially, the chicks had high DVoc levels, showing an anxiety characteristic. The results also revealed that *S. tortuosum* extracts displayed significant anti-depression activities [69]. *Sceletium* plants have been reported to have calming effects and improve mood. The phytochemical profile of *S. tortuosum* has been isolated and has identified psychoactive compounds. Different experimental models and clinical studies have proved that *Sceletium* plants reduce anxiety and promote calmness.

### 5.5. Analgesic Properties of Sceletium Plants

Native Khoi and San tribes of South Africa have been traditionally using “Kanna” plants as a pain reliever by chewing the plant material directly and smoking the residue after chewing [12,66,70]. However, recently, this chewable plant has been prepared in various forms such as capsules, gel caps, resin, teas, and tinctures, which can be used as a snuff and smoked [70,71,72]. Despite the extensive documented history of the psychological and biological effect, few studies have proved that the *Sceletium* plant possesses useful analgesic properties [10,28,66,70]. The effects of *S. tortuosum* were investigated in male Sprague-Dawley rats. The results revealed that mesembrine, as a major compound in *S. tortuosum*, can be responsible for analgesic activity [66]. A previous study conducted by Dimpfel and co-workers [12] showed the statistical significance of theta wave reduction. In accordance with delta, beta1 and alpha2 are distinct features of analgesic drugs, although such studies have not been extensively carried out.

Moreover, Zembrin promoted the EEG changes usually shown in analgesic properties of standardized drugs [12]. Smith et al. [70] examined the psychoactive compounds of *Sceletium* plant and reported bioactivities such as relief of discomfort and pain (analgesia) [70]. A study was also conducted on the role of CREB signaling in Alzheimer’s disease and other cognitive disorders using an electroencephalogram. The results revealed that the alkaloid mesembrine decreased the θ waves, which reduced the δ, α2, and β1, affecting analgesic activity [73]. Modern painkillers are highly addictive, toxic, and ineffective due to the development of tolerance. For this reason, medicinal plants such as *S. tortuosum* are gaining in popularity since they are natural analgesics, which have been used for centuries [12].

### 5.6. Anti-Inflammatory Properties of Sceletium Plants

Inflammation is basically the body’s response to microbial infection. Thus, white blood cells or leukocytes release a chemical into the body, which, in turn, increases the blood flow in the context of tissue injury. Moreover, this process serves the prevention of further spreading to the rest of the body as protection [74]. Bennett and colleagues [13] reported that the cytokine was associated with inflammatory stimulus.

Hence, cytokine response affected the interleukin-6 (IL-6) and monocyte chemotactic protein-1 (MCP-1), with *p* < 0.005 and *p* < 0.0005, respectively. The results showed high concentrations of mesembrine *Sceletium* extracts, demonstrating anti-inflammatory properties [13]. In vivo studies, male Wistar rats were administered *S. tortuosum* extracts. The results showed significant comprehensive anti-inflammatory effects of *Sceletium* extracts [10]. Cytokinin profiles of *S. tortuosum* were investigated and the results showed anti-inflammatory activities [30]. Indigenous *S. tortuosum* plants have been reported to possess various biological properties including psychoactivity, anti-inflammatory, anti-oxidant, immunodulatory, antidepressant, anxiolytic as well as promoting a sense of well-being. Studies have attributed it to the countless high level of alkaloids (Table 1).

The literature revealed that different techniques can be used to determine the biological activities of *Sceletium* sp. Furthermore, different models such as in vitro culture studies, animal studies as well as human studies are available in the current review. Table 2 provides a summary of the different methods used to determine the biological properties of *Sceletium* sp.

Several studies and historical knowledge of medicinal plants including *S. tortuosum* suggest that the plant has endless downstream opportunities [83,84,85,86]. However, there are quality control [85,86,87,88] and safety [89,90] issues when dealing with medicinal plants. Similar to any production or manufacturing, Good Manufacturing Practice (GMP) regulations must be adhered to, which incorporate evaluating and monitoring aspects to promote standard quality and safety of products, for example, plant medicine. There is inconsistency in terms of the composition of plant extracts since they are harvested from nature, possess varieties of genetics, ecological, and environmental variances, hence, the quality will differ [91]. In addition, another main concern in the usage of medicinal plants is identifying the bioactive compounds, which can be attributed to the biological properties displayed by the medicinal plant [92].

Several researchers have investigated the quality of medicinal plants [93] in South African by using analytical quality control techniques such as NMR, HPTLC [94], UHPLC-MS [95,96], HPLC [97] and GC-MS [98]. There is a need to focus on new technological advances such as metabolomics high-throughput quantification to detect low molecular weight compounds.

This should also serve in various industries such as the medical, pharmaceutical. and veterinary sectors due to the abundant therapeutic properties. Regardless, the current review has shown that very few studies have focused on this plant. To date, there are promising studies that need expansion and big data analysis. Innovative approaches are essential in order to obtain natural novel bioactive compounds from *Sceletium* sp. for the exploration of biological and psychological effectiveness. More recently, Zhao and co-workers [27] studied nuclear magnetic resonance (NMR) metabolomics of *S. tortuosum*, which is a powerful technique to obtain accurate quantitative and qualitative data. Furthermore, this method bypasses chromatographic separation, thus the complete metabolite information is available [27]. There are several advantages of the method such as unbiased fingerprinting, accurate, low limitation of detection, generation of big data, high throughput, quantitative, low-molecular weight molecules, and in vivo and vitro studies. Several applications including toxicity, mutants, response to environmental stress, nutrition, diseases, and drug development can be extrapolated from the analysis of metabolomics [99].

## 6. Conclusions and Future Prospects

Nowadays, plant-based medication or/and remedies have become a global growing trend, therefore increasing the demand for natural bioactive compounds from natural sources. Thus, researchers foresee tremendous challenges surrounding various aspects of plant-based medicine. Although South Africa has recognized African Traditional Medicine as part and parcel of the public health care system, there are serious shortcomings such as the quality, standardization, efficacy, authenticity, and safety of plant-based medicine. Current biotechnological advances focus on genetic transformation and metabolic engineering techniques as innovative approaches to enhance the in vitro production of secondary metabolites in medicinal plants. Thus, this will lead to novel metabolites for the drug development process. There is need for in-depth studies to extract and identify bioactive compounds in the biological and pharmaceutical investigation of medicinal plants, particularly *S. tortuosum.* Due to the economic state of South Africa, a vast majority of the population (27 million people or 52%) use traditional medicine as an alternative for their primary health care requirements. Affordability, availability, and indigenous knowledge of medicinal plants make them an ideal solution to the overpriced, toxic conventional medications on the market. Prehistoric evidence supported by preliminary studies show that the *S. tortuosum* plant exhibits useful bioactive compounds for anti-depressant and anxiolytic activity, thus promoting a sense of well-being. Toxicological assays are a requirement to determine the safety parameters of *S. tortuosum*.

In this review, we examined the uncharted territories of *S. tortuosum* research, which will spark important studies in niche areas of chemistry, medicine, and natural products. This should also serve in various industries such as the medical, pharmaceutical, and veterinary sectors due to their abundant therapeutic properties. Moreover, several studies have been carried out in vitro as well as animal and clinical studies to prove the potency and efficacy of *S. tortuosum*. With regard to the abundant bioactivity displayed by *S. tortuosum*, “Kanna” functions as a “Chewable Cure” to promote a sense of well-being.

## Figures and Tables

**Figure 1 molecules-26-02557-f001:**
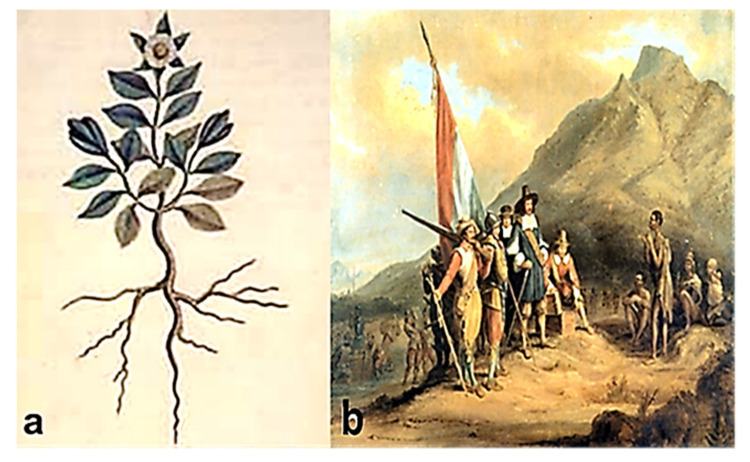
*Sceletium* painting done by Simon van der Stel in 1685 (**a**), Jan van Riebeek with his Dutch colleagues at the Cape of Good Hope (**b**) [16].

**Figure 2 molecules-26-02557-f002:**
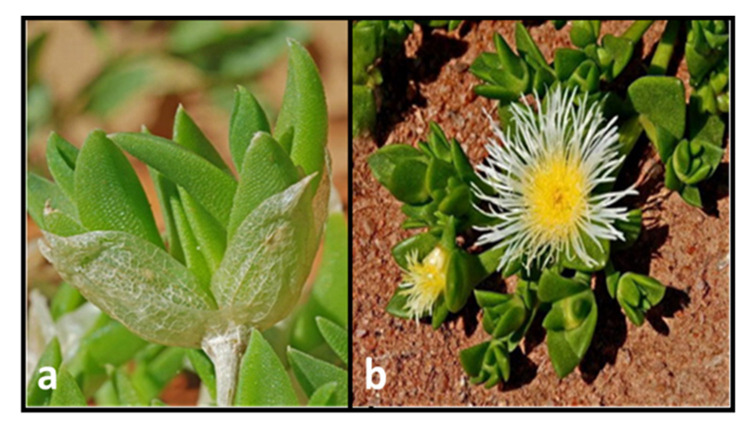
Dried out leaves of *Sceletium tortuosum* (**a**) and bright yellow flowering plant (**b**) [1].

**Figure 3 molecules-26-02557-f003:**
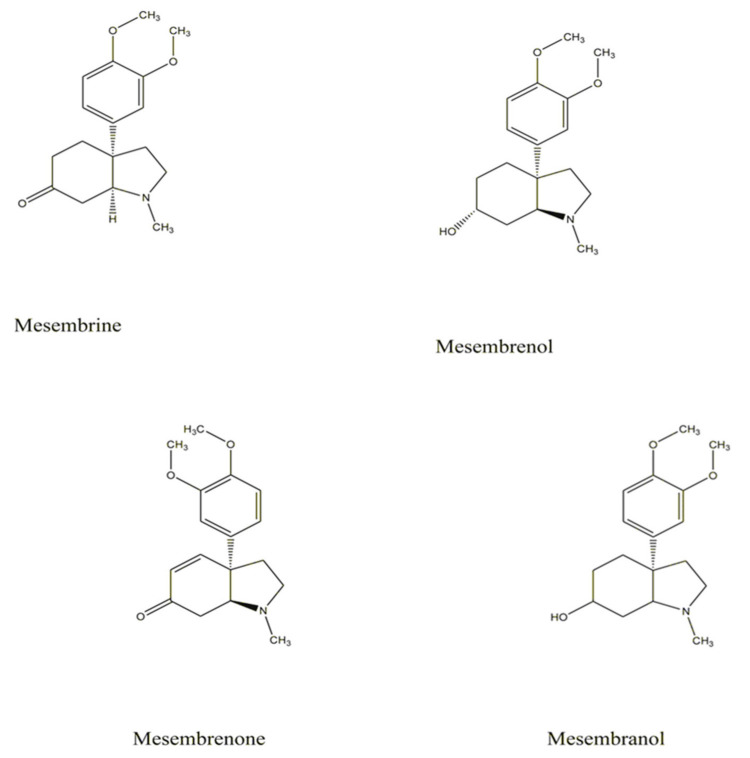
Chemical structures of the mesembrine alkaloids of *Sceletium tortuosum* [24,29].

**Table 1 molecules-26-02557-t001:** Biological investigation of Sceletium plants.

Biological Activity	Part Used	Chemical Composition	Ref.
Antidepressant-like properties, anxiolytic, mood elevator	Not specified	Alkaloids	[1]
Toxicological tests	*Sceletium tortuosum* (Zembrin^®^)	Mesembrenone, mesembrenol and mesembrin	[8]
Anxiety, depression	Whole plant	Alkaloids	[9]
Antioxidant and anti-inflammatory, neuroprotective effects	Leaves	Mesembrine	[13]
Immunomodulatory effects, depression, inflammation	*Sceletium tortuosum* (Trimesemine™)	Alkaloids	[28]
Antiviral, antioxidant activity	Whole plant	Anthraquinones, terpenes, polyphenols, anthocyanin, tannins, alkaloids, glycosides, carbohydrates and coumarins	[36]
Sleep function, improves memory and enhances cognitive function	Whole plant	Alkaloids	[43]
Anxiolytic effect	*Sceletium tortuosum* (Zembrin^®^)	Alkaloids	[48]
Central nervous system activity	Whole plant	Alkaloids	[61]
Antimalarial activity	Whole plant	Mesembrine	[75,76,77]
Stress-related illnesses	Whole plant	Δ7-mesembrenone, mesembrenone and mesembrine.	[78]
Immunomodulatory effects	leaves	Δ7-mesembrenone	[79]
Neurocognitive effects	*Sceletium tortuosum* (Zembrin^®^)	Alkaloids	[80]
Antidepressant	Not specified	Mesembrines, Mesembrenol, mesembrenone	[59]
Safety, tolerability, promotes coping with stress and sleep.	*Sceletium tortuosum* (Zembrin^®^)	Mesembrine	[81]
Toxicological tests	Whole plant	Mesembrine, mainly the O- and N demethyl-dihydro, hydroxy, and bis-demethyl-dihydro metabolites	[82]

**Table 2 molecules-26-02557-t002:** Methodology of biological properties of *Sceletium* sp.

Aim of the Study	Methodology Used			Ref.
	Plant Preparation	Biological Techniques	Phytochemical Screening	
To determine psychological effects of *Sceletium tortuosum* in an in vivo model	*S. tortuosum* parts were ground into powder form.	In vivo model, using Male Wistar rats in a double placebo study. Oral administration of extract for 17 days and stress was induced. Behavior was monitored.	N/A	[10]
To investigate anti-HIV and free radical scavenging activity of crude extracts of *S. tortuosum.*	*S. tortuosum* parts were ground into powder form. Crude extracts of *S. tortuosum* prepared using ethanol and ethyl acetate.	Anti-HIV assays FRET-based Sensolyte HIV-1 kit used. Fluorogenic substrate mixed with extracts in a 96 well plate. Fluorescence intensity measured. DPPH assay Extract mixed in ethanol and the absorbance measured.	Total phenolic content Total flavonoids content Total tannin content	[36]
To determine the acute effects of *Sceletium tortuosum* (Zembrin) in human amygdala	Sceletium (Zembrin)	Sceletium (Zembrin) administrated to humans in a placebo double-blind, cross-over study.	N/A	[48]
To screen for anxiolytic activity of *Sceletium tortuosum*, using an in vivo zebrafish model	*S. tortuosum* parts ground into powder form. Crude extracts of *S. tortuosum* prepared in various solvents.	MTC assay used in different concentrations in the zebrafish model study. Larvae placed in a 48-well plate with the control. MTC plates incubated for 18 h at 28.5 °C.	N/A	[60]
To determine the effects of *Sceletium tortuosum* in the Chick Anxiety-Depression Model	*S. tortuosum* parts ground into powder form and water used as a solvent.	Preclinical model assay on Male Silver Laced Wyandotte chick administrated *S. tortuosum* in various concentrations and induced with stress. Distress vocalizations captured the data.	N/A	[68]

## Data Availability

Not applicable.

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
