# Peer review of "A Chewable Cure “Kanna”: Biological and Pharmaceutical Properties of Sceletium tortuosum"

_molecules, 2021, doi:10.3390/molecules26092557_

Round 1

Reviewer 1 Report

This is a review article describing various aspects of Sceletium tortuosum and summarizing its biological and pharmaceutical properties as the major subject. In a developing country such as South Africa, this plant is widely and traditionally utilized for the treatment in clinical anxiety and depression, psychological and psychiatric disorders, also improving mood and promoting relaxation and happiness. Although there were a few simple mistakes, this article seemed well documented and orderly organized, and also seemed to be clear and easy to understand. Even the subject was quite specific and therefore would be interesting only in a particular region of traditional herbal medicine, but this review would be able to make a considerable contribution to the progress of knowledge in this field.

Regarding English, the section 1 to 3 of this manuscript seemed well-written except for some minor faults, but the latter section might have so many issues as described below. Particularly, the structure of sentence seemed weird and improper in many parts, and it would be necessary and important to solve these problems by making these sentences more simple and clear. Also, the authors should pay attention to punctuate appropriately long sentences for making its easy to read and understand. There were so many points against a grammatical rule, and these must be corrected. Curious to say, the authors always put a comma behind the reference numbers in a square bracket. It was totally questionable why. This appeared to separate the sentences, and therefore the meaning of these sentences might be more difficult to understand.

To sum up, this manuscript was considered to be absolutely incomplete and insufficient. The structures of sentences, the wordings and the expressions should be entirely revised, and all grammatical errors should be corrected. When revising the manuscript, it would be absolutely necessary to get the advice of a native speaker.  

(Points should be revised)

  • Line 30:The phrase “From the” seemed quite strange, and its meaning was totally unclear and not understandable. It might be, so to speak, the removal remainder of sentence, probably due to a mistake to erase former sentence.
  • Line 62: The phrase “phosphodies- terase4”might be a typing error, and it would probably be “phosphodiesterase4”.
  • Line 172: The preposition “on”seemed unnecessary, and it should be deleted. Also, it would be better to change the word “potential” to a plural form “potentials”.
  • Line 175: The word “alkaloid”should be a plural form “alkaloids”. Also, the phrase “the dried the plant material” had simple errors, and it should be “the dried plant materials”.
  • Lines 175-176: The phrase “there is allows aquestion” seemed hardly understandable, and the word “allows” seemed especially hard to understand.
  • Line 176: The phrase “the phytochemicals compounds”was a simple mistake, and it would be “the phytochemical compounds”.
  • Lines177-180: This part “Traditional practices demonstrated that Sceletium plants has abilities to elevate mood, reduce stress, tension, anti-anxiety and its tranquilizing properties [35] and also in the treatment of abdominal pain, toothache, some people chew, smoke or use as tea or snuff mostly for pressure [15]”. The structure of sentence was unnatural and awkward, and it also seemed easy to be confused. Therefore, the revision of this part would be recommended. One example of revision was follows: “The traditional practices demonstrated that Sceletium plants had the abilities for elevating mood and reducing stress and tension, of anti-anxiety and tranquilizing properties [35], and were used for the treatment of abdominal pain and toothache. Also, some people chewed, smoke, drink as tea, or snuff it mostly for relieving pressure [15]”.
  • Line 180: The word “Current,”seemed weird a little, and it would be better to say “Today,” or “Nowadays”.
  • Line 182: The phrase “such as in vitro, animal models”should be “such as an in vitro and in vivo animal models”.
  • Lines 183-185: The structure of sentence seemed weird and confused, because of improper or excessive use of punctuation. It would be better to divide this sentence in two between the words “potential”and “furthermore” on line 184.
  • Lines 185-186: The word “produced”seemed not appropriate, and it would be better to say “contained”.
  • Lines 190-191: The sentence “the outcomes showed dose-dependent rich source of antiradical activity phytochemicals using scavenge DPPH radicals”seemed quite chaotic and incomprehensible. It would be completely unclear what the meaning might be, even not imaginable what the authors intended to say here. Therefore, it would be absolutely necessary to be entirely corrected.
  • Line 192: The word “lead”should be “leading”.
  • Line 193: The phrase “for the discovery of”seemed quite awkward, and it might also be unnecessary, and therefore would be possible to be replaced with “as”.
  • Line 197: The preposition “to”should be “in”.
  • Line 198: The phrase “a key factor”seemed odd, and it would probably be “ a key role”.
  • Lines 201-202: The phrase “with serious implication to”seemed weird, and its meaning was unclear.
  • Line 203: The phrase “approximately 77% of individual who are stressed,”seemed incomplete, and the preposition “in” might be necessary. Then, the phrase would be better to say ”in approximately 77% of individual who are stressed,”.
  • Line 206: The phrase “Various formulations”might be wrong, and it should be “Various formulae” or “formulas”.
  • Line 208: The subject was a plural form, and therefore the verb “support”was wrong. It should be “support”.
  • Line 209: The phrase “Current research emphasis on”seemed wrong, and it would be “Current researches emphasize”.
  • Lines 214-217: This was the phrase to explain thein vivo experimental system, but not a complete sentence (lines 214-216), and therefore this might be combined with the following sentence “Half of the test rats were subjected to moderated stress not exceeding 1 h for the last 3 days of treatment (lines 216-217).
  • Line 231: The relative pronoun “which”seemed unnecessary, and it should be deleted.
  • Line 268: The phrase “global disability worldwide”seemed quite weird, and the meaning was not understandable. The word “global” seemed inappropriate, and the true meaning of the word “worldwide” was not questionable.
  • Line 270: It might be better to replace the word “Individual”with “Individuals”.
  • Lines 271-272: The phrase “Sharma and co-workers [60],review showed” seemed odd and unnatural. It seemed questionable why putting a comma “,” behind [60] and why placing “review” and “showed”.
  • Line 272: The conjunction “that”should be deleted. Or “ that” should be followed by a clause, but the phrase following “that” did not have a verb, and hence it was not a clause. The structure of this sentence was quite odd, might be wrong.
  • Line 275: The phrase “Maphanga et al. [60],”was totally not understandable.
  • Line 277: The word “conducted”was wrong, and it should be “were subjected to”.
  • Lines 277-278:The phrase “was determined using 5-day post-fertilization (dpf) zebrafish larvae”made no sense, because the subject of this phrase was completely unknown, or might not be given here.
  • Line 279: The word “anti-anxiety”was used in a wrong manner, and it would be “the anti-anxiety activity” or “the anti-anxiety effect”.
  • Line 280: The meaning of the word “reiterate”seemed not understandable.
  • Line 283: The phrase “possess agonist of GABAA,”seemed unnatural and inappropriate, because mesembrine could not possess any receptor. It would be better to say “might be the agonists of receptors” or “had the agonist actions on the receptors”.
  • Line 285: The word “human”seemed superfluous, and it should be deleted.
  • Lines 293-294: The phrase “the anxiety-depression effectiveness”seemed quite awkward, and it would be better to say “the effectiveness against anxiety and depression”.
  • Line 295: The phrase “induced with”seemed inappropriate, and it would be good to say “exposed to”.
  • Line 297: The word “prior”would be better to say “prior to stress”.
  • Line 306: The phrase “research studies”seemed odd, and it would be “researches” or “studies”.
  • Line 307: The sentence “Anin-vivo study investigated the effects of tortuosum in male Sprague-Dawley rats” was wrong, because any study could investigate nothing. It would be better to say “The effects of S. tortuosum were investigated in male Sprague-Dawley rats”. In addition, this sentence was still lacking the object of “effects”.
  • Line 308: The verb “demonstrates”should be a past form “demonstrated”.
  • Line 309: The phrase “have been”would be better to be replaced with “might be”.
  • Lines 312-313: Same to the line 307, the sentence “although such studies have not been extensively investigated”was wrong, and it would be better to say “although such studies have not been extensively carried out.
  • Lines 317-318: The phrase “using electroencephalogram”was wrong, and it should be “using an electroencephalography”.
  • Line 321: It seemed unknown whether the phrase“external microbial infection” might be proper or not, because it could not be found in any dictionary.
  • Line 322: The phrase “the body’s”seemed unnecessary, and should be deleted.
  • Line 324: The phrase “serve as protection to prevent further spreading to the rest of the body”might have a meaning different from the intention of authors, and it would be better to say “serve the prevention of further spreading to the rest of the body as protection”.
  • Lines 324-326: The sentence “Bennett and colleagues [13], reported two tortuosumplant extracts were prepared using different extraction processes” seemed not suitable and not necessary at all in this part. Frankly to say, it was nonsense and should be deleted.
  • Lines 327-328: The sentence seemed incomplete, and the word “significant”might be wrong. It was unclear what of IL-6 and MCP-1 might be affected by cytokine response.
  • Line 332: The phrase “Sreekissoon et al. [75],”was quite strange, and it might be totally not understandable why here. This phrase was completely far from other part of the sentence. Might be meaningless as it was.
  • Line 333: The word “display”seemed weird, and it might be used to expose something to public view. Therefore, it would be better to replace “display” with “showed”.
  • Lines 334-335: The word “inflammatory”would probably be “anti-inflammatory”.
  • Line 336: The phrase “have be attributed to”was wrong, and it might be “have been attributed to”.
  • Line 339: It might be necessary to insert a hyphen between “plant”and “based”.
  • Line 342: It might be necessary to insert a hyphen between “plant”and “based”, same to line 339.
  • Line 345: Same to the comments No. 53 and 54.
  • Line 346: The phrase “adhere to”might be wrong, and it would probably be “adhered”.
  • Lines 354-360: This part seemed extremely redundant and absolutely unnecessary to recount all techniques here. It would be good enough to describe like “NMR, HPLC and so on [97-101]”.  
  • Line 360: The phrase “Future prospects are focusingon” seemed quite odd , and would probably be wrong. Because the word “prospect” is an uncountable noun, this phrase would therefore be like “Future prospect was to focus on”.
  • Line 364: The word “Due”should be “Due to”.
  • Lines 371-372: The phrase “several studies have investigated”seemed strange, because the studies investigated nothing. It should say “several studies have been carried out in”.

Reviewer 2 Report

The manuscript is very well written and easy to read. Is a topic of interest and deals with an unknown plant, at least to me, linked to a specific area and culture. There are not many papers published on it (I have found 38 in PubMed), so no sure it is enough for a review.

The list of medicinal plants, its ethnobotany, uses, and applications are huge. The benefit from mot of them is claimed in the current literature, but I have missed, in the present review, a comparison with other species.

A review like this, should contain a methodological section, in which approaches, and techniques be summarized. Is this species investigated by using, for example, the modern -omics techniques? In vitro culture techniques should also be mentioned, as an alternative to field frown plants.

Should the investigation on this species take a biotechnology direction? Which one?

The paper is quite descriptive, based on the literature survey. There is more future than past in the research so the final section should be developed more in detail.

Reviewer 3 Report

Dear authors,

The review manuscript entitled " A Chewable Cure “Kanna”: Biological and pharmaceutical properties of Sceletium tortuosum” provides an extensive overview of the biological and pharmaceutical properties of Sceletium tortuosum as well as the bioactive compounds with emphases on antimicrobial, anti-inflammatory, anti-oxidant, antidepressant, anxiolytic and other significant biological effects.

It presents scientific relevance for the area of Chemistry, medicine and natural products area. After consulting www.sciencedirect.com, all authors all authors have published articles, with antimicrobial and / or similar studies. The language (English) are satisfactory (I suggest the final revision)!

However, you need to change some details/informations in the “abstract”, “Introduction” and “conclusions”. I request information on methodological design for obtaining information.

Abstract: Adequate, but I suggest:

- Page 1, lines 14, 25, 20, 25…: I suggest rewriting the species or genus name, in italics!

- In Keywords, I suggest inserting other words, such as: biological properties and / or bioactive compounds!

  1. Introduction section: It is well written, but:

- I suggest rewriting the species or genus name, in italics, in the introduction and throughout the entire manuscript.

- Page 2, lines 56-75: The paragraph is too long! I suggest dividing it!

- At the end of this section, I suggest inserting information about the methodological design for the writing of this review, such as (descriptors and databases used, criteria for inclusion and exclusion of the analyzed articles, period of consulted publications, etc.)!

  1. History, description and distribution of Sceletium: It is well written, but:

- I suggest improving the quality of figure 1.

- Page 3, lines 111-132: The paragraph is too long! I suggest dividing it!

  1. Biological and pharmaceutical properties of Sceletium sp.: It is well written, but:

- Page 6 (lines 212 – 232); Pages 6-7 (lines 234 – 265); Pages 7-8 (lines 267 – 299); Page 8 (lines 301 – 319): The paragraphs are too long! I suggest dividing them!

- I suggest expanding the discussions in the subsections: “4.1. Antimicrobial properties of Sceletium plants”; “4.5. Analgesic properties of Sceletium plants” and “4.6. Anti-inflammatory properties of Sceletium plants”.

- At the end of the section, I suggest inserting table 1 (cited on pages 9-10) and writing 1 or 2 paragraphs, contextualizing the proposal of the manuscript.

  1. Conclusion and Future prospects

- Generally, “references” are not mentioned in the "conclusion" section! I suggest using part of the quoted text to expand the discussions, in the previous sections.

- Generally, “tables” are not mentioned in the "conclusion" section! In addition, this table is loose! It was not cited in the text!

- I suggest moving table 1 to the end of the " Biological and pharmaceutical properties of Sceletium sp." section.

- At the end of the section, I suggest highlighting the importance of the review for Chemistry, medicine and natural products area!

* Tables and Figures: Adequate! Please, see proposed suggestions for Figure 1!

* References: Please, check if the references are in accordance with the journal's rules.

Round 2

Reviewer 1 Report

As mentioned before, English seemed extremely poor, and therefore it might be reasonable to consider that this manuscript could not yet be ready to submit for publication. Actually, there were too many points required for revision or correction in this manuscript, and they would be too numerous to mentioned here. So, just a part of the revised points was described below. At long last, it would be strongly recommended to exhaustively revise whole article with the aid of native speaker. Again, it must be necessary to emphasize that the most important point was to ask a native speaker of English for help.

For example,

  1. Page 1, lines 39-40: The sentence seemed awkward, and it would be better, probably clear to say “practiced by incorporating current Western medical practice withtraditional medical healthcare”.
  2. Page 1, line 42: The phrase “healthcare needs”seemed inappropriate, and it would be good enough to say “healthcare” in case of using “utilized” as a verb.
  3. Page 1, line 42: The phrase “Besides that this forms part of”seemed awkward, probably wrong, and it should be better to say “Besides it forming a part of”.
  4. Page 1, line 43: The phrase “traditional healthcare system”might require an article “the”.
  5. Page 2, line 45: The phrase “in both the private as well as the public healthcare system”was completely wrong, and a preposition should be “by” or “with”. In addition, it could be acceptable to use “both” and “as well as”
  6. Page 2, line 48: The word “worldwide”would be better to be replaced with “around the world” or other word in this case.
  7. Page 2, line 49: The preposition “with” should be “in”.
  8. Page 2, lines 49-50: The sentence “South Africa is a country rich with traditional healing practices, diverse fauna and flora”seemed quite unnatural and awkward, because two phrases “healing practices” and “diverse fauna and flora” were separated just by a comma. It would be better to say “South Africa is rich in traditional healing methods and diverse fauna and flora”.
  9. Page 2, line 51: The word “contribute”seemed unpopular, and it would probably be better to say “account for”.
  10. Page 2, lines 55-59: It seemed questionable whether “The bioactive compounds such as alkaloids, phenolic, flavonoids, tannins, glycosides, saponins, and terpenoids compounds with tremendous therapeutic health benefits”might be a sentence or not. In fact, the revision of the original sentence was possibly considered to make the original much worse by dividing it into two sentences. The second half was totally wrong, and could not make any sense. Perhaps, it seemed possible to consider that the original manuscript was supposed to be revised without understanding sufficiently and adequately the reviewers’
  11. It would be necessary to emphasize that the use of a punctuation mark was inappropriate, and should be more careful to apply a punctuation, especially comma, to separate the sentences at where in the manuscript.

and many others.
